# Investigation on the Mechanical Properties and Strengthening Mechanism of Solid-Waste–Sulfur-Based Cementitious Composites

**DOI:** 10.3390/ma16031203

**Published:** 2023-01-31

**Authors:** Jiaxin Liu, Changwang Yan, Jie Li, Ju Zhang, Shuguang Liu

**Affiliations:** 1School of Civil Engineering, Inner Mongolia University of Technology, Hohhot 010051, China; 2School of Mining, Inner Mongolia University of Technology, Hohhot 010051, China; 3Engineering Research Center of Inner Mongolia Autonomous Region for Ecological Building Materials and Assembly Structure, Hohhot 010051, China

**Keywords:** sulfur-based cementitious composites, solid waste, mechanical properties, strengthening mechanism

## Abstract

This research used waste ceramic powder (CP) to replace aggregate, fly ash (FA) as filler, and combined them with sulfur to prepare composite cementitious materials. The variations of the mechanical properties with the aggregate proportions (aggregate mass/total mass) of 65%, 70%, and 75%, and the FA contents (FA mass/aggregate and filler mass) of 0%, 10%, 20%, 30%, 40%, and 50% were studied. The correlation evaluation model of sulfur content, CP content, FA content, and mechanical properties was established using the gray correlation theory, and the comprehensive mechanical property evaluation model was established as the foundation of the entropy method. Finally, the optimum proportion of the solid-waste–sulfur-based cementitious composites was determined. Results showed that, without FA, the CP increased from 65% to 75% and the comprehensive mechanical properties of the specimen increased by 60.53%. After FA was added, the peak point of the comprehensive mechanical properties appeared in group S75F10, which was 0.9210. During the hardening of the cementitious material, sulfur was mainly used as a binder, CP played the role of skeleton and part of the filler, whereas, as a crystal nucleus, the FA promoted the transformation of the sulfur crystals. Both the CP and FA can reduce the porosity of the specimen to a certain extent and have potential defect repair ability, thus densifying the matrix and improving the strength. When the proportion of sulfur: CP: FA is 1:2.7:0.3, the flexural (FS), compressive (CS), and splitting tensile (STS) strengths of the specimen are 14.8, 86.2, and 6.8 MPa, respectively. The flexural (FCR) and tensile (TCR) compression ratios are 0.172 and 0.079, respectively.

## 1. Introduction

The annual output of sulfur is roughly 70 million metric tons [1], which is stored in pyrite, coal, crude oil, and natural gas. The combination of sulfur-containing waste gas with water in the air causes acid rain, which directly harms people’s living environment. Thus, in order to comply with environmental protection laws and regulations, oil or natural gas refineries all over the world have relatively complete desulfurization facilities [2]. This phenomenon makes the current global total production of elemental sulfur 10–20% more than the global aggregate demand and is expected to continue to increase in the future [3]. Numerous experiments are carried out and used in various fields, such as hydrogen production from sulfur [4], sulfur rubber [5], and sulfur building materials [6], to consume industrial sulfur on a large scale. Sulfur concrete (SC) prepared using a modified sulfur polymer as a binder is used in different working conditions to replace cement concrete and has excellent performance, such as rapid hardening, corrosion resistance, no seasonal restriction, and recyclability. Furthermore, SC has been used in road pavements [7], hydraulic structures [8], and retaining walls [8]. In addition, compared with Portland cement, the emissions produced by SC are low, and the heat released from the mixing process of SC manufacturing (about 120 °C) is lower than the calcination temperature of cement clinker (about 1450 °C) [9].

Given the thermal expansion and contraction of sulfur, the volume of sulfur reduces during the hardening process of SC. At room temperature, the stable crystal system of sulfur is orthorhombic sulfur (Sα), which has a volume shrinkage rate of 12%. However, the nonuniform shrinkage of sulfur pores and high contraction stress during the hardening process due to the uneven temperature distribution of fresh slurry remarkably reduces the mechanical performance of SC [10]. Currently, the modification of sulfur is a useful method for decreasing volume contraction. With this technique, the sulfur crystal can be changed into a monoclinic sulfur crystal (S_β_), which has a minimal volume contraction (7%) [11]. Nevertheless, the mechanical characteristics of SC are also reduced by the 7% volume contraction of modified sulfur. Therefore, replacing sulfur with some filler is considered to diminish the pores formed after the sclerosis of SC and improve the compactness of the matrix. Fly ash (FA) is a commonly used filler, and its addition can considerably promote the leaching rate and mechanical properties of SC [8,12]. Additionally, research in academia is concentrated on employing solid waste materials rather than natural aggregate to create green concrete because doing so has obvious advantages for the environment, the economy, and society [13]. Waste-ceramic-recycled aggregate has decreased rigidity but greater porosity and water absorption in contrast to natural aggregate [14]. Brito et al. [15] simply crushed waste ceramics and used them as coarse aggregate. The prepared concrete has low strength and can only be applied to nonweight-bearing structures. Anderson et al. [16] tested the effect of 25–100% waste ceramic content on the mechanical characteristic of recycled concrete and discovered that the mechanical characteristic of recycled concrete with ceramic are almost the same as those of ordinary concrete. Nepomuceno et al. [17] used waste ceramics with the identical granule size as a substitute for natural sand to prepare concrete. The flexural (FS), compressive (CS), and splitting tensile (STS) strengths of recyclable concrete decrease to different degrees but meet the normal-use requirements. Tortikul et al. [18] crushed and screened waste ceramics into fine aggregate, which proved the practicality of using ceramics as fine aggregate to prepare mortar. Binici et al. [19] used broken waste ceramics as fine aggregate to substitue natural river sand to prepare recycled concrete and showed that, when the replacement rate of waste ceramics is 40–60%, its CS, erosion resistance, and durability improved. Therefore, based on modified sulfur, the collection and processing of waste ceramics into powder to substitute nonrenewable river sand completely as fine aggregate and their mixing with FA filler to prepare solid-waste–sulfur-based cementitious composites (WSCCs) to achieve improved mechanical properties and replace traditional cement are feasible.

An accurate understanding of the relationship between the basic mechanical performance of cementitious composites and the ratio of various materials is the basis for further research. The gray correlation theory [20] can take an uncertain system as the research object to establish a correlation degree model and reflect the effect degree of the corresponding factor sequence in accordance with the gray correlation coefficient of the data sequence, which can locate the solution of the complicated hassle with the lack of information. Many scholars have used this method for auxiliary research. Zhu et al. [21] established a forecast model of the CS of concrete with recycled aggregate according to the gray correlation analysis. Cui et al. [22] also predicted the CS of concrete containing slag and metakaolin by the extreme gradient enhancement method based on the gray correlation evaluation. Mokhtar et al. [23] and Jin et al. [24] considered that the employment of the gray relational principle to discuss the affecting factors and changing rules of concrete strength is feasible. Zhang et al. [25] extended the gray correlation theory to the research of CS and micropore structural parameters. In addition to the research on the mechanical properties, Zhang et al. [26] successfully determined the optimal mix ratio of superfine-cement-based mud on the basis of the Taguchi gray correlation analysis. Kong et al. [27] applied the gray correlation model to the sensitivity assay of the effect of aggregate on the interface transition zone and put forward a new suggestion on the optimal option of aggregate when preparing well-performing concrete with the enhanced interface transition zone. In addition, considering that the mechanical properties of the specimen have multiple indices, its comprehensive mechanical properties and the optimum dosage of the three influencing factors are impossible to determine using a single index. Therefore, a perfect evaluation system is needed when selecting the optimal ratio. The entropy procedure is a weighting method based on target changeability [28] which can lessen the mistakes between dissimilar assessment indices and is extensively utilized in assessment systems in miscellaneous domains. Chen et al. [29] put forward the flood disaster evaluation index by the entropy analysis of the normalization factors of historical flood data. Sahoo et al. [30] characterized the water quality based on a variety of water quality indicators and in combination with the Bayes’ rule through the index weight calculated by the entropy method. Mi et al. [31] combined the entropy method with the variation theory and established an evaluation system of coal mine safety by using the collected 17 evaluation indices. Yao et al. [32] derived a fuzzy entropy multicriteria risk evaluation model for hydropower stations by quantifying the uncertainty in fuzzy sets. In the field of architecture, Gong et al. [33] established an extensive assessment model of magnesium oxychloride cement concrete following the entropy method and provided an evaluation procedure of the crucial level of its extensive water opposition factor. Qin et al. [34] determined the weight of the mass, CS, and STS of an FA fiber-reinforced concrete specimen by the entropy method and put forward a method to assess the comprehensive strength of the resulting concrete with longevity value as an assessment index. Thus, the variation law of the durability value is imitated, and the evolution law of concrete durability under diverse working situations is predicted. Therefore, the comprehensive mechanical properties of each group of specimens are determined by the entropy method after collecting several mechanical properties of specimens, and the best dosages of sulfur, CP, and FA in reference to the comprehensive mechanical characteristics of specimens are selected is scientific.

In summary, the methods to enhance the mechanical characteristics of sulfur cementitious materials mostly focus on a single modification or addition of filler. However, the improvement of the mechanical characteristics of modified sulfur cementitious materials by combining the resource advantages and gain effects of waste ceramics and FA, making waste ceramics into aggregate to completely replace natural river sand, and relying on the synergistic gain effect of CP and FA, should be investigated. The principal purpose of this paper is to offer a neoteric sustainable method for the enhancement of the mechanical characteristics of a sulfur-based cementitious composite by introducing CP to replace natural aggregate completely and sulfur and FA filler for the preparation of WSCC. This study also aims to increase the dosage of CP and FA as much as possible to achieve the best dosage without sacrificing the mechanical strength. Compared with cement-based concrete and traditional SC, the proposed strategy may have numerous advantages, i.e., less carbon dioxide emission, lower life-cycle cost, and excellent mechanical properties. This investigation can supply a theoretical reference for the employment of WSCC in practical projects, and the whole process conforms to the concept of sustainable development.

## 2. Experimental Design

### 2.1. Raw Materials

The industrial sulfur, modifier dicyclopentadiene (DCPD), and FA used in the experiment are shown in Figure 1. Industrial sulfur was obtained from China Anqing Guoxing Chemical Co., Ltd. (Anqing, China), yellow granular, with a melting point 119 °C, and insoluble in water. DCPD from Shanghai Flax Technology Development Co., Ltd (Shanghai, China). was selected as the modifier. DCPD is an organic compound, a colorless crystal, and insoluble in water. DCPD is manufactured by China Shanghai Flax Technology Development Co., Ltd. The CP used in the experiment was produced by Hunyuan Hong Jun New Material Co., Ltd (Datong, China). The particle size distribution of the CP is referred to the Chinese GB/T 17671-1999 [35], as shown in Table 1, and the preparation process is shown in Figure 2. The collected waste ceramics were cleaned and put into a drying box at 105 °C for 6 h. The ceramics were initially broken with a jaw crusher. Then, they were further fabricated into powder with a disk prototyping machine. The ceramic powder was sieved according to the standard particle size and the ceramic powder fine aggregate was obtained. The apparent density and silt content referred to the Chinese GB/T 14684-2011 [36]. The main physical performance of waste ceramics is displayed in Table 2. The chemical composition is displayed in Table 3. The FA used in the experiment is Class I high-calcium FA produced by Dalate Banner Thermal Power Plant in Erdos, China, the particle size distribution is displayed in Figure 3, and the chemical constitution of the FA is shown in Table 4.

### 2.2. Specimen Preparation

In the preliminary test, our team found that, when the content of fine aggregate was 60%, the cementing material floated up during the vibration process, leading to the stratification of the specimen. When the content of fine aggregate is 80%, there are many defects in the specimen prepared due to insufficient cementing material which cannot wrap the fine aggregate. In addition, Margareth et al. [37] found in the test that the content of fine aggregate increased from 50% to 80%, the amount of sulfur decreased from 50% to 20%, and the CS of the specimen increased from 12.12 MPa to 27.07 MPa and then decreased to 14.48 MPa. Among them, the peak point of the compressive strength occurs in the range of 60–80% fine aggregate content. The experimental conclusion of Gwon et al. [38] showed that the content of fly ash as a filler was generally less than 50%. Based on the above reasons, the sulfur, CP, FA were weighed in accordance with the mixture ratio shown in Table 5.

In accordance with the Chinese standard [35], the dimension of the SWCC sample was 40 mm × 40 mm × 160 mm and the oven temperature was 170 °C. The mold was brushed with a special oil for demolding and placed into the oven together with the aggregate and filler. When the oven temperature increased to 170 °C, the 135 °C oil bath pot was started to heat the sulfur. When the sulfur was completely melted, the aggregate and filler were collected from the oven and slowly poured into the oil bath pot, stirred, and mixed with a handheld stirrer 20 min. The mold was then placed on a vibrating table. The stirred slurry material was poured into the mold while vibrating for 5 s, and the surface of the mold was scraped with a flat shovel to finish the vibration. After pouring, the mold was cooled at 20 ± 2 °C for 2 h before being demolded, then cured in the laboratory. Considering the thermoplasticity of the sulfur-based composites, no hydration reaction occurred during the entire manufacturing process [38,39,40], so theoretically no special curing time was required. The sulfur-based material can obtain 70–80% compressive strength within 24 h [41]. In reference to the curing time of the sulfur-based material specimens by Moon et al. [40], the mechanical property test was conducted after 24 h curing in this paper.

### 2.3. Test Methods

#### 2.3.1. FS Test

Specimens were tested to comply with the FS test method [35]. The testing machine adopted was the 60 T uniaxial pressure testing machine (YAS-600) and the loading rate was 2400 ± 200 N/s. The three vertical planes passing through the three cylindrical axes should be parallel, and continue to be parallel, and equidistant perpendicular to the test body during the test. One of the supporting and loading cylinders can be slightly tilted so that the cylinder and the test body are fully in contact so that the load is evenly distributed along the width of the test body without any torsional stress. The FS was calculated using Formula (1).
(1)$$R_{f}=\frac{1.5F_{f}L}{b^{3}}$$
where *R_f_* is the FS (MPa), *F_f_* is the load implemented to the center of the prism when it is broken (N), *L* is the length between the supporting prism (100 mm), and *b* is the side length of the square section of the prism (40 mm).

#### 2.3.2. CS Test

Specimens were subjected to the CS test method [35]. The testing machine adopted was the 60 T uniaxial pressure testing machine (YAS-600) and the loading rate was 2400 ± 200 N/s. When loading, the piston of the press should be perpendicular to the center of the specimen and perpendicular to the longitudinal axis of the machine. This ensures that the resultant force of the loading force passes across the central of the specimen. The load plate of the press is formed perpendicular to the axis of the machine and remains unchanged in the loading process. The center of the ball seat of the pressing plate on the press shall be at the intersection point of the vertical line of the machine and the underside of the upper pressing plate, and its tolerance is soil 1 m. The top load plate can be readjusted automatically when it is in contact with the test body, but the position of the top and underside load plate should be fixed during loading. The CS was calculated in accordance with Formula (2).
(2)$$R_{C}=\frac{F_{C}}{A}$$
where *R_c_* is the CS of mortar (MPa), *F_c_* is the load implemented to the center of the prism when it is broken (N), and *A* is the compressive area (mm^3^).

#### 2.3.3. STS Test

In accordance with the STS test method [42], the testing machine adopted the 60 T uniaxial pressure testing machine (YAS-600). The specimen is placed in the center of the pressure plate under the testing machine, and the splitting pressure surface and the wild crack surface should be perpendicular to the top surface of the specimen forming: between the upper and lower pressure plate and the specimen pad with a round strong pad and a pad strip, where the pad and the pad strip should be aligned with the center line of the specimen on and below and perpendicular to the top surface when forming. The STS was calculated using Formula (3).
(3)$$f_{ts}=\frac{2F}{\pi A}$$
where *f_ts_* is the STS (MPa), *F* is the breaking load (N), and *A* is the splitting region of the specimen (mm^2^, *A* 1600 mm^2^).

#### 2.3.4. Rapid Air 457 Test

The specimen was cut into 40 mm × 60 mm × 20 mm slices and the surface was ground, polished, blackened, and covered with a coat of zinc oxide. The pore size distribution, specific surface area, gas content, stomata number, and stomata spacing of the specimen were tested using the RapidAir 457. Figure 4 shows the working principle of the RapidAir 457. With reference to the linear-traverse method [43], take any straight line in the hardened sample, and the ratio of the total length of the line segment intercepted by the stomata on the straight line to the full length of the line is the volume content of the stomata in the sample. Formulas (4)–(8) are the calculation formulas of the stomata parameters, such as air content, average chord length, and specific surface area. Average chord length of the stomata:(4)$$\bar{l}=\frac{\sum l}{n}$$
where $\bar{l}$ is the average chord length of the stomata, mm; ∑l is the total chord length of the stomata cut by the whole wire, mm; *n* is total number of stomata cut by the whole conductor.Specific surface area:(5)$$\alpha =\frac{4}{\bar{l}}$$
where α is the specific surface area, mm^2^/mm^3^.Mean stomata radius:(6)$$r=\frac{3}{4}\bar{l}$$
where *r* is the mean pore radius, mm.Air content:(7)$$A=\frac{\sum l}{T}$$Number of pores in the 1000 mm^3^ sample:(8)$$n_{v}=\frac{3A}{4\pi r^{3}}$$
where *n_v_* is the number of pores in the 1000 mm^3^ sample.

#### 2.3.5. NMR Test

The specimen was cut into 40 mm × 40 mm × 50 mm specimens, which were immersed with water for 24 h in vacuum saturation equipment. The MesoMR—60 S NMR instrument was used to analyze the aperture composition of the specimen. The magnet in the instrument is a permanent magnet, the magnetic field intensity is 0.50 ± 0.8 T, and the main frequency is 21.3 MHz. The variation of the NMR *T*_2_ spectrum can reflect the variation of the stomata structure of the porous media. According to the principle of the NMR, the distribution of the NMR *T*_2_ spectrum is related to the aperture, and this process can be simplified as [44,45]:(9)$$\frac{1}{T_{2}}=\rho_{2}{\left(\frac{S}{V}\right)}_{\mathrm{Porosity}}$$
where *T*_2_ is the free relaxation time of the pore fluid, ms; *ρ*_2_ is the transverse surface relaxation strength of *T*_2_, μm/ms; *S* is the stomata surface area, cm^2^; *V* is the fluid volume in the pore, cm^3^.For samples that can be equivalent to spherical and cylindrical holes, Equation (9) can be further converted into the relationship between the *T*_2_ relaxation time and aperture [46]:(10)$$\frac{1}{T_{2}}=\frac{\rho_{2}}{\gamma_{c}}F_{s}$$
where *γ_c_* is the radius of the hole; *F_s_* is the geometric factor, spherical hole *F_s_* = 3, and cylindrical hole *F_s_* = 2.Then, Formula (10) can be adjusted as:(11)$$r_{c}=CT_{2}$$
where *C* = *ρ*_2_
*F_s_*, *C* is the transformation coefficient.

It can be seen from Formulas (9)–(11) that the pore radius is proportional to *T*_2_ and the distribution of *T*_2_ can mirror the pore dimension data [47].

#### 2.3.6. SEM Test

The core pieces of the specimen were selected after the load experiment, and the specimen was sprayed with gold by using an ion-sputtering instrument, and its microstructure was observed with the Hitachi S-4800 electron microscope (Hitachi Limited, Tokyo, Japan). The secondary electron imaging resolution was 1.0 nm at 15 kV and the accelerating voltage was 0.1–30 kV. The effects of the sulfur, CP, and FA in the hardening process of the specimens can be further understood by observing the micromorphology of the samples.

### 2.4. Model Methods

The gray relational evaluation is a vital section of the gray gadget theory. This technique locates the numerical relationship among subsystems in the gadget via positive methods. The fundamental thought of the gray relational evaluation is to infer the relationship between unique sequences via the geometric kingdom of the principal sequence curve and the similarity between subsequences [48]. The diploma of correlation refers to the dimension of the correlation between the elements of two sequences, which will alternate with time or distinctive objects. During the development of a system, if the altering vogue of the two sequences is regular or the level of synchronous trade is high, the correlation between the two sequences is high. Otherwise, the correlation between the two sequences is low.

The entropy weight method is a method to resolve the objective weight of the response [49]. According to the principle of probability theory, the entropy of different responses can be calculated. This determines the importance of each response in decision-making, which mainly includes determining the target matrix, standardizing processing, calculating the probability of the response occurrence, calculating the entropy of the response, calculating the deviation of the response, and the weight value [50]. Combined with the gray correlation method and EWM, the optimal ratio of the raw materials in this experiment can be determined.

## 3. Results and Discussion

### 3.1. Mechanical Properties of WSCC

#### 3.1.1. Flexural Strength (FS)

The mean values of the FS values for three specimens in each group are shown in Figure 5. The FS of the specimens increases significantly with the increasing aggregate proportion. Compared with that of the S65 group, the FS values of the S70 and S75 groups increase by 10.10% and 21.23%, respectively. The FS increases by about 10% when the aggregate content is 5%. Similar conclusions are also obtained from Lopez et al. [51] by displacing part of the natural sand of concrete with CP, and increasing the CP replacement levels from 0% to 40% resulted in a 14.67% increase in the FS of the specimen.

In addition, the FS increases initially and then declines with the increasing dosage of the FA. The FS of each group reaches the maximum when the FA content is 10%. Taking the FS of the three groups without FA as the benchmark strength, when the dosage of FA is enhanced from 0% to 50%, the FS values of the S65, S70, and S75 groups change from 12.28% to −12.28%, from 5.34% to −20.61%, and from 5.71% to −14.29%, respectively. Huang et al. [52] also found that the FS of concrete increases from 28 days to 365 days after partial cement is displaced by FA, which is attributed to the formation of recrystallized calcium carbonate in the cemented matrix caused by the reaction between FA and calcium hydroxide. This phenomenon reduces the porosity of the matrix and transition zone. However, in the present experiment, the sulfur material does not undergo a hydration reaction [40]. The excellent FS is speculated to be due to FA, which reduces the porosity.

Therefore, the FS of the specimen can be improved by appropriately increasing the amount of aggregate, and the FS increases evidently after adding a proper amount of FA filler. However, the FS of specimens with more than 20% FA content is lower than that without FA. When the amount of aggregate is 75% and the amount of FA is 10%, the FS of the specimen is highest.

#### 3.1.2. Compressive Strength (CS)

The CS values of each group of specimens measured by the test are shown in Figure 6. Under the same amount of FA, the CS values of the S65 and S75 groups are lowest and highest, respectively, except for the three groups of specimens without FA. Taking the CS of the S65 group as the benchmark strength, the CS values of the S70 and S75 groups are increased by 2.51% and 5.03%, respectively. According to the experimental phenomenon of Lopez et al. [51], the present experiment found that using CP instead of sand as a fine aggregate enhanced the CS of the specimen remarkably. Given this phenomenon, Chen et al. [53] believe that, apart from being a fine aggregate, CP acts as a filler to reduce the gap between particles of cementitious materials, thus playing a role in enhancing the force of the dense matrix.

With the enhancement of the dosage of FA, the CS of different series specimens increases initially and then gradually decreases. In specimens with 0% FA, the CS of the S70 specimens is moderately greater than the S75 specimens. Taking S65 specimen increases as an example, when the content of FA in specimens is enhanced from 10% to 50%, the CS decreases from 15.42% to −5.48% compared with that of the specimens without FA. Martinovic et al. [54] and Gwon et al. [55] considered that the modified sulfur causes curing shrinkage due to temperature change, leading to microcrack damage. Shin et al. [8] obtained the same conclusion by conducting CS tests on the SC specimens with FA and found that the CS of the specimen increased by 39% with FA replacement levels of 14.5%.

Therefore, when only CP is taken as aggregate in the WSCC, the CS of the specimen increases until the content of CP reaches 70%. However, if FA is added as the filler, the CS of the specimen can be further improved. The consequence of FA on the CS of WSCC is not remarkable as that on the FS.

#### 3.1.3. Splitting Tensile Strength (STS)

STS values are shown in Figure 7. The STS of the specimens increased with the increased aggregate proportion. Taking the STS of S65 group specimens as the benchmark strength, the STS values of S70 and S75 groups increase by 9.12% and 20.32%, respectively. STS increases by about 5.03% when the ceramic aggregate content increases by 5%. Lopez et al. [51] also found that using CP can improve the STS of specimens. When the amount of ceramic powder is increased from 10% to 40%, the STS of the sample increased by 12.12%.

When the dosage of FA rises, the STS values of the three groups of specimens with different aggregate ratios increase initially and then decrease. Taking the STS of specimens without FA as the benchmark strength, when the content of FA changes from 10% to 50%, the STS values of the S65, S70, and S75 groups change from 11.63% to −20.93%, from 8.00% to −24.00%, and from 8.00% to −24.00%, respectively. Shin et al. [8] had a similar conclusion, i.e., when replacing modified sulfur with 14.5% FA, the STS of the specimen increased by 83.3%.

Therefore, the STS of the specimens can be slightly increased by appropriately increasing the amount of CP. If 10% FA filler is used in conjunction with the aggregate, the STS of the specimens can be remarkably improved. In addition, the adverse consequences of FA on the STS of three groups of specimens with different aggregate ratios do not show a consistent rule. The FA contents of three groups of specimens with STS lower than the initial value were 40%, 20%, and 30%, respectively.

#### 3.1.4. Flexural (FCR) and Tension (TCR) Compression Ratios

The ratio of the FS to the CS of the specimen is the FCR, which can characterize the fracture performance of the specimen. A high FCR indicates improved fracture performance of the specimen. The ratio of STS to CS is the TCR. As one of the indices to measure the mechanical performance of specimens, the TCR can reflect the relative development speed of the two strengths and characterize the brittleness of the matrix material. Figure 8 shows the variation law of the FCR and TCR of three groups of specimens with different aggregate ratios when the amount of FA is different.

When the aggregate proportion increases, the FCR of the specimen increases significantly. The FCR of the specimen with 75% CP is 0.032 higher than that with 65% CP. After adding FA, the changing trend of the FCR of the three groups of specimens is slightly different. The FCR values of the S65 and S70 groups first decrease and then increase, whereas the FCR curve of the S75 group shows a “W” shape. However, the three groups of specimens show a common feature, that is, the supplement of FA has no significant positive effect on the FCR. Compared with those of specimens supplemented with FA, the fracture performance of specimens with only CP as aggregate is better.

Unlike the curve of the FCR, the curve of the TCR shows another characteristic. When the matrix only has CP, the TCR increases with ceramic powder content. The TCR of the specimen with 75% CP content is 19.35% higher than that of the specimen with 65% CP content. After the addition of FA, when the dosage of FA is 0–10%, the TCR of the S75 group significantly improved, reaching 0.079, whereas those of S65 and S70 groups decreased. When the dosage of FA is 10–40%, the TCR of the S75 group decreases evidently (by 25.32%). The TCR of the S65 group decreases slightly but not significantly (by 5.00%), but that of the S70 group decreases first and then rises again. When the FA level exceeds 40%, the TCR values of the S65 and S70 groups drop again, whereas that of the S75 group rises slightly. When the aggregate is 75% and FA filler is 10%, the brittleness of the specimens is ameliorated best.

According to the above tests, the FS, CS, and STS of the WSCC are improved most evidently when the level of FA filler is 10%. Thus, the surface morphology of each group of specimens of 0% and 10% FA is observed and shown in Figure 9. When the matrix is not filled with FA, the surface roughness and delamination of the sample are evident with increased CP content. At this time, although the mechanical properties of the specimen are improved, the freshly mixed slurry in the test process is difficult to vibrate and pour, and the workability of the matrix is relatively poor. However, after adding 10% FA, the surface of the specimen becomes smooth and dense, and this improvement becomes increasingly evident with an increasing amount of aggregate. The reason why the combination of CP and FA improves the appearance and mechanical strength of the specimen needs further microexperimental research to determine.

### 3.2. Microstructure and Strength Formation

#### 3.2.1. Pore Distribution and Strength Formation

The specimens of the S70 and S75 groups with a high aggregate content are selected for microscopic study. The bright-green area in Figure 10 shows the stomata distribution of the specimens. According to the linear-traverse method [43], any straight line is selected in the sample, the pore diameter refers to the length of the line segment that the straight line passes through the pore, and pore number refers to the number of pores that the straight line passes through. Figure 11 is a summary of the test results from the linear wire method showing the number and proportion of stomata with different stomata sizes. The abscissa of the bar graph represents the size of the stomata, and the ordinate represents the number of stomata with different size. The pie chart shows the proportion of pores of different sizes in the sample. Figure 12 demonstrates the gas content, average chord length of the stomata, and specific surface area. Air content is the proportion of the volume of the stomata in the test sample to the total volume of the sample. The larger the air content, the more stomata in the sample. The average chord length reflects the change of stomata size inside the sample. The larger the average chord length is, the larger the stomata and the larger the proportion of harmful stomata in the sample. Stomatal specific surface area refers to the stomatal surface area per unit volume in the sample. The larger the specific surface area is, the more complex the inner surface roughness of the stomatal is; correspondingly, the fewer the number of large stomata, the larger the number of small stomata [56].

When increasing the aggregate content from 70% to 75%, the bright-green area in Figure 10 increases sharply, which means that the number of pores in the specimen is increasing. After adding 10% FA filler, the bright-green area decreases remarkably, signifying that the FA filler can reduce the number of pores in the specimen. Little difference is observed in the bright-green area between the S70 and S70F10 groups. When the aggregate content is 70%, the proportion of sulfur increases correspondingly, and the cementing material is enough to ameliorate the pore defects of the specimens. Thus, the impact of adding the FA filler is not significant enough.

When FA filler is not added, the S75F0 group has a decreased number of 0–100 μm pores, a decreased number of 0–30 μm pores (by 6.42%), evidently an increased number of >100 μm pores, and an increased number of >500 μm pores (by 1490.48%) compared with that of the S70F0 group. When the aggregate has only CP, increasing the amount of CP remarkably increases the pore size of the specimens. After adding 10% FA filler, the proportion of 0–30 μm pores of S70F10 and S75F10 specimens increase, but the proportion of pore > 30 μm decreases remarkably. Gwon et al. [38] thought that using FA usually reduced the general porosity of the SWCC. As the total pore volume decreased, the CS increased, which is in agreement with the strength–porosity relationship [57]. In addition, several studies suggested that the strength increase by using FA may be due to the decreased small holes, whereas the existence of large holes did not seem to affect the CS of sulfur composites [58].

Using 10% FA filler can reduce the air content and average chord length of the pores in the S75 group by 62.71% and 63.24%, which are higher than those of the S70 group (by 26.26% and 18.52%, respectively). Using 10% FA filler can increase the specific surface area in the S75 group by 173.69%, which is higher than that in the S70 group by 23.98%. A high air content of the specimen and long average chord length of pores results in the introduction of increased large-diameter spherical bubbles. The existence of these bubbles enhances the porosity of the SWCC, diminishes the density of the SWCC, and carries off the diminution of the effective carrying capacity section of the specimen. When the specimen is stressed, the stress concentration easily occurs near the pore, which results in reduced mechanical performance. This problem can be effectively avoided by adding FA filler.

#### 3.2.2. *T*_2_ Spectrogram and Strength Formation

The *T*_2_ spectrum is a time constant signifying the recovery process of the transverse component of the nuclear magnetization, also known as the transverse relaxation time, which is caused by the exchange of energy within the nuclear spin system [59]. The shorter the relaxation time, the smaller the porosity of the specimen, and the longer the relaxation time, the larger the porosity of the specimen. The total area of *T*_2_ spectrum can signify the porosity of the specimen, and the peak area can represent the number of pore sizes within the corresponding size range [60].

Figure 13a displays the variation trend of the *T*_2_ spectrum curve of the S70 group. Three peaks correspond to micro-, medium-, and large-radius pores in the specimens. The envelope area of the *T*_2_ spectrum curve of S70F10 is smaller than that of S70F0, and the relaxation time of peak 3 of the *T*_2_ spectrum curve moves forward. This result indicates that adding 10% FA filler can efficaciously lessen the number and volume of pores, and the improvement effect for macropores is significant. Figure 13b demonstrates the *T*_2_ spectrum of the S75 group, and the spectral line of S75F0 has only two peak points. The envelope area of peak 5 is large, which indicates that only increasing the CP content without adding FA filler increases the medium and large pores in the specimens. S75F10 is observed with three peaks, and the medium and large pores are remarkably reduced.

The existence of pores leads to the stress concentration of the specimen when it is stressed, and if the number of pores is extremely large, the bearing capacity of the specimen is affected. However, 10% FA filler, which fills the natural defects of sulfur and improves the grading of CP, can well alleviate this problem, thus reducing the pore defects caused by an insufficient sulfur dosage. On the basis that the mechanical properties of the specimen have been improved by a high content of CP, the compactness of the specimen is further strengthened to improve the bearing capacity. Gwon et al. [38] also established that the improvement of the mechanical characteristics of specimens by FA may be related to the reduction of the porosity.

#### 3.2.3. Microappearance and Strength Formation

At room temperature, the crystal stable form of sulfur is orthorhombic (Sa), and other allotropes include monoclinic sulfur (S_β_) and polymeric sulfur (S_∞_), in which Sa and S_β_ are composed of sulfur eight rings (S_8_), while S_∞_ is a long chain structure with 10^6^ atoms. S_γ_ is the state of sulfur at the temperature of 159 °C, and all sulfur is composed of S_8_ at this time. When the temperature is higher than 159 °C, the sulfur is in the S_μ_ state, and the sulfur is in the mixed state of S_8_ and S_∞_ molecules.

S75F0 and S75F10 are selected for microscopic observation (Figure 14). As can be seen from Figure 14a, most of the modified sulfur (S_β_) in the hardened slurry of sulfur-based materials is composed of plate-like single crystals, and a small part of the modified sulfur is polymerized into a chain structure. Juhyuk et al. [40] thinks that this change is the result of unsaturated hydrocarbon in the organic modifier decomposing the S_8_ ring of the liquid sulfur and polymerizing it to form a long chain structure, and the sulfur with the long chain polymer structure has better durability. Nevertheless, some defects can be clearly seen in the matrix with a high ceramic aggregate content, and the surface also presents a honeycomb flocculent structure, which indicates that the bonding effect between the aggregate and matrix is not ideal.

The spherical particles of the FA coated with sulfur can be observed in Figure 14b, and an evident chemical reaction trace in its appearance is not observed. The flocculent substances that exist in S75F0 have almost disappeared completely, and defects are remarkably reduced, leaving only the microcracks at the bonding place of the matrix. According to the analysis, the main reason for the formation of microfractures in the contact conversion area of the specimen is that the slurry is stirred at 135 °C and is directly placed in a room-temperature environment to cool down. The surface of the specimen is in direct contact with air and a steel mold preheated at 170 °C. The surface of the specimen solidifies first, but the internal heat dissipation slowly forms a temperature difference, which leads to the asynchronous shrinkage of the volume between the cementitious material and aggregate, and finally forming microfractures in the contact conversion area.

The cooling crystallization mechanism and microscopic morphology of sulfur shows that the FA is physically bonded with the sulfur; that is, the FA particles are enfolded and constant through the liquid sulfur. Nonhydrated FA granules may also have the function of crystal kernels during sulfur crystallization [41]. In the system of sulfur chilling and crystallization, the FA particles act as the kernel of the sulfur crystal and promote crystal transformation [61]. Liquid sulfur accumulates and radically changes the direction of the spatial trajectory towards the FA particles, which makes the sulfur crystals develop and enlarge in the vicinity of the FA particles and cohere into a compact matrix. The addition of FA replaces the phase of sulfur. Thus, the wide variety and extent of pores shaped through sulfur shrinkage, and the shrinkage stress in the WSCC, are diminished. In addition, the FA particles have a large contact area and a thickening influence on the grout body. The stickiness of the grout body blended with the FA makes the distribution of CP steady throughout the stirring of the molding. It has been proved that properly covering the fine aggregate and fly ash particles with modified sulfur may be an important factor for the formation of discrete pores and the uniform and firm structure of the matrix [40]. The exposed ceramic powder aggregate and fly ash filler can be used as the weak interface transition zone of the WSCC. The ideal covering state can be realized by changing the mixing ratio of raw materials and mixing conditions (i.e., mixing speed and temperature).

## 4. Correlation Analysis and Mixture Proportion Determination of SWCC Based on Mechanical Properties

### 4.1. Correlation Evaluation Model

Given that the dosage of sulfur, CP, and FA affect the mechanical properties of the specimens, instituting a model to evaluate the correlation between the influencing factors and mechanical properties is indispensable. The FS, CS, STS, FCR, and TCR of the specimens are different under different proportions of CP and FA. Therefore, the gray correlation technique is chosen to determine the relationship of the dosages of sulfur, CP, and FA with FS, CS, STS, FCR, and TCR. The contents of sulfur, CP, and FA are chosen as evaluation columns, and the FS, CS, STS, FCR, and TCR are chosen as reference columns. The technique of deciding the gray correlation model between three influencing elements and five mechanical overall performance indices is as follows:

The initial matrix is constructed as follows:(12)$$\left(X_{1},X_{2},\cdots ,X_{n}\right)=\left(\begin{array}{cccc} x_{1}\left(1\right) & x_{1}\left(2\right) & \cdots & x_{1}\left(n\right)\\ x_{2}\left(1\right) & x_{2}\left(2\right) & \cdots & x_{2}\left(n\right)\\ \vdots & \vdots & \ddots & \vdots \\ x_{m}\left(1\right) & x_{m}\left(2\right) & \cdots & x_{m}\left(n\right) \end{array}\right),\;\mathrm{and}$$
(13)$$\left(Y_{1},Y_{2},\cdots ,Y_{n}\right)=\left(\begin{array}{cccc} y_{1}\left(1\right) & y_{1}\left(2\right) & \cdots & y_{1}\left(n\right)\\ y\left(1\right) & y_{2}\left(2\right) & \cdots & y_{2}\left(n\right)\\ \vdots & \vdots & \ddots & \vdots \\ y_{m}\left(1\right) & y_{m}\left(2\right) & \cdots & y_{m}\left(n\right) \end{array}\right).$$

In the matrix of (*X*_1_, *X*_2_, …, *X_n_*), *m* = 18 and *n* = 3. In the matrix of (*Y*_1_, *Y*_2_, …, *Y_n_*), *m* = 18 and *n* = 5.

The initial matrix is averaged as follows:(14)$$X_{ij}^{\prime }=\frac{X_{ij}}{{\bar{X}}_{j}}=\left(\begin{array}{cccc} \frac{x_{1}\left(1\right)}{\bar{x\left(1\right)}} & \frac{x_{1}\left(2\right)}{\bar{x\left(2\right)}} & \cdots & \frac{x_{1}\left(n\right)}{\bar{x\left(n\right)}}\\ \frac{x_{2}\left(1\right)}{\bar{x\left(1\right)}} & \frac{x_{2}\left(2\right)}{\bar{x\left(2\right)}} & \cdots & \frac{x_{2}\left(n\right)}{\bar{x\left(n\right)}}\\ \vdots & \vdots & \ddots & \vdots \\ \frac{x_{m}\left(1\right)}{\bar{x\left(1\right)}} & \frac{x_{m}\left(2\right)}{\bar{x\left(2\right)}} & \cdots & \frac{x_{m}\left(n\right)}{\bar{x\left(n\right)}} \end{array}\right)\;\;i=1,2,3\cdots ,m;j=1,2,3\cdots ,n.$$
(15)$$Y_{ij}^{\prime }=\frac{Y_{ij}}{{\bar{Y}}_{j}}=\left(\begin{array}{cccc} \frac{y_{1}\left(1\right)}{\bar{y\left(1\right)}} & \frac{y_{1}\left(2\right)}{\bar{y\left(2\right)}} & \cdots & \frac{y_{1}\left(n\right)}{\bar{y\left(n\right)}}\\ \frac{y_{2}\left(1\right)}{\bar{y\left(1\right)}} & \frac{y_{2}\left(2\right)}{\bar{y\left(2\right)}} & \cdots & \frac{y_{2}\left(n\right)}{\bar{y\left(n\right)}}\\ \vdots & \vdots & \ddots & \vdots \\ \frac{y_{m}\left(1\right)}{\bar{x\left(1\right)}} & \frac{y_{m}\left(2\right)}{\bar{y\left(2\right)}} & \cdots & \frac{y_{m}\left(n\right)}{\bar{y\left(n\right)}} \end{array}\right)\;\;i=1,2,3\cdots ,m;j=1,2,3\cdots ,n.$$

The difference matrix is calculated as follows:(16)$$\Delta_{ij}=\left(\begin{array}{cccc} \Delta_{1}\left(1\right) & \Delta_{1}\left(2\right) & \cdots & \Delta_{1}\left(n\right)\\ \Delta_{2}\left(1\right) & \Delta_{2}\left(2\right) & \cdots & \Delta_{2}\left(n\right)\\ \vdots & \vdots & \ddots & \vdots \\ \Delta_{m}\left(1\right) & \Delta_{m}\left(2\right) & \cdots & \Delta_{m}\left(n\right) \end{array}\right)\;\;i=1,2,3\cdots ,m;j=1,2,3\cdots ,n.$$
$$\Delta_{ij}=\left|Y_{ij}^{\prime }-X_{ij}^{\prime }\right|$$

The maximum and minimum differences are determined as follows:(17)$$\left\{\begin{array}{c} V_{\max}=\max\Delta_{ij}\\ V_{\min}=\min\Delta_{ij} \end{array}\right.$$

The correlation coefficient of the gray entropy (*γ_ij_*) is calculated as follows:(18)$${ϒ}_{ij}=\frac{V_{\min}+\xi V_{\max}}{\Delta_{ij}+\xi V_{\max}}$$

*ξ* is the resolution coefficient and 0.5 in this paper.

The gray correlation degree (*G*) is calculated using the following equation:(19)$$G=\frac{1}{m}\overset{m}{\underset{i=1}{\sum }}{ϒ}_{ij}\;\;i=1,2,3\cdots ,m;j=1,2,3\cdots ,n.$$

### 4.2. Correlation Evaluation of Influencing Factors and Indicators

The average processing results of Formulas (14) and (15) are displayed in Table 6, and the calculation effects of the gray correlation degree are proven in Figure 15. In accordance with the calculation effects in the figure, the gray correlation degree between influencing elements and reference targets is sorted. The gray correlation degree between influencing elements and FS is CP > sulfur > FA. The gray correlation degree between influencing elements and CS is sulfur > CP > FA. The gray correlation degree between influencing elements and STS is CP > sulfur > FA. The gray correlation degree between the influencing elements and the FCR of the specimen is CP > sulfur > FA. The gray correlation degree between the influencing elements and the TCR is CP > sulfur > FA.

The CP content has a high correlation with the four indices of the specimen (i.e., FS, 0.8353; STS, 0.8080; FCR, 0.8480; TCR, 0.8109). The correlation degree between sulfur dosage and CS is also significant, reaching 0.8121, and the correlation degrees between the sulfur dosage and FS, STS, FCR, and TCR are slightly lower than CP but is maintained between 0.75 and 0.80. The correlation degree between the FA content and five mechanical performance indices is maintained at about 0.50. The dosages of CP and sulfur are considered to play an important role in the FS, CS, STS, FCR, and the TCR of specimens, and FA also has a certain gain effect. CP plays the role of the skeleton, sulfur plays a role in bonding and lubrication, and FA plays the role of filling, and the three components influence one another. However, the optimum mixture ratio needs to be further discussed.

### 4.3. Comprehensive Mechanical Performance Evaluation Model

The comparison of the test results of the mechanical properties of each group of specimens shows that the FS, CS, STS, and TCR of the S75F10 group are the best, but the FCR is not the best. Therefore, the ratio of sulfur, CP, and FA in the S75F10 group of specimens cannot be considered to produce the best comprehensive mechanical properties. The gray entropy principle is picked out to give weight to the five mechanical properties indices of the specimens for the determination of the comprehensive mechanical properties and omnifaceted evaluation of the mechanical properties of each group of specimens. Then, in accordance with the order of the results of the comprehensive mechanical properties, the optimum mixture ratio is selected. The evaluation system of the comprehensive mechanical performance index is expressed as follows:(20)$$S=t_{1}S_{1}+t_{2}S_{2}+t_{3}S_{3}+\ldots t_{n}S_{n}=\sum_{i=1}^{n}t_{i}S_{i}$$
where *t_i_* is the weight of the target and *S_i_* is a standardized target.

The FS, CS, STS, FCR, and TCR of the specimen are arranged in turn as the original data matrix, and the initial sequence matrix is still the matrix in Formula (13): (*Y*_1_, *Y*_2_, …, *Y_n_*), where *m* = 18 and *n* = 5.

The original matrix is processed to obtain the maximum response of various mechanical properties.
(21)$$C_{i}\left(j\right)=\frac{y_{i}\left(j\right)-\min\left(y_{j},j=1,2,3\cdots ,n\right)}{\max\left(z_{j},j=1,2,3\cdots ,n\right)-\min\left(z_{j},j=1,2,3\cdots ,n\right)}\;i=1,2,3\cdots ,m$$

The reference matrix is constructed as follows:(22)$$C^{*}=\left(C_{1}^{*}\;C_{2}^{*}\cdots C_{5}^{*}\right)=\left(1.0000\;1.0000\;1.0000\;1.0000\;1.0000\right).$$

The difference matrix ($\Delta_{ij}^{\prime }=|C^{*}-C_{i}\left(j\right)|$) between the reference and initial matrices is calculated in accordance with Formula (16). In the constructed new matrix, the maximum and minimum values are selected in accordance with Formula (17) and the new correlation coefficient (*γ_ij_*′) is calculated in accordance with Formula (18). Then, a weight is given to each mechanical performance index, and the process is as follows.

The proportion of each index (*P*) is determined as follows:(23)$$P_{ij}=\frac{{ϒ}_{ij}^{\prime }}{\sum_{i=1}^{m}{ϒ}_{ij}^{\prime }}\;\;i=1,2,3\cdots ,m;j=1,2,3\cdots ,n.$$

Entropy (*e*) is determined as follows:(24)$$e_{j}=-\frac{1}{\ln\left(m\right)}\sum_{i=1}^{m}P_{ij}\ln\left(P_{ij}\right)\;j=1,2,3\cdots ,n.$$

The weight (*W*) of every target is computed as follows:(25)$$W_{j}=\frac{1-e_{j}}{\sum_{j=1}^{n}\left(1-e_{j}\right)}\;j=1,2,3\cdots ,n.$$

The comprehensive mechanical properties of each group of specimens are determined as follows:(26)$$K_{i}=\sum_{i=1}^{m}\Delta_{ij}^{\prime }W_{j}\;j=1,2,3\cdots ,n.$$

### 4.4. Determination of the Mixture Proportion Based on Comprehensive Mechanical Properties

Figure 16 shows the comprehensive mechanical properties of all specimens. Under the different proportions of aggregate and filler, the complete mechanical traits of the specimens improve at first and then reduce with the extent of the quantity of FA. Three peak points are observed in the comprehensive mechanical properties of the specimens, which, respectively, correspond to those of the S65, S70, and S75 groups with 10% FA content. This result indicates that this quantity of FA filler has a remarkable effect on mechanical properties. Comparing the comprehensive mechanical properties of 18 groups of specimens, the S75F10 group is the best followed by the S75F0 and S70F10 groups.

In summary, the optimum dosage of FA filler is 10% for the WSCC with waste CP as aggregate. In this experiment, the comprehensive mechanical properties of the S75F10 group are the best. The optimum mixture ratio of sulfur: CP: FA is 1:2.7:0.3. When the proportion of sulfur: CP: FA is 1:2.7:0.3, the FS, CS, and STS of the specimen are 14.8, 86.2, and 6.8 MPa, respectively. At this point, the synergistic effect of sulfur–CP–FA has reached an ideal state, and the mechanical bite force between particles is largest. The freshly mixed slurry in the construction process is easy to pour and vibrate, minimal defects in the specimen are observed after cooling and molding, and the mechanical traits of the specimen can be remarkably enhanced to meet the actual engineering needs.

## 5. Conclusions

In this study, on the basis of modified sulfur, WSCC is prepared using waste CP as aggregate and adding FA filler. We investigate the effect of the content of CP and FA on the mechanical traits of the specimens and discuss its microstructure and strength formation mechanism. In addition, on the basis of the gray theory, two research models are established, and the influences of sulfur, CP, and FA on the mechanical properties are analyzed. The mixture ratio of the WSCC has also been scientifically evaluated. The principal discovery of this research is as follows.When the dosage of the aggregate is 75% and the dosage of FA filler is 10%, the FS, CS, and STS of the WSCC increase most evidently. Increasing the proportion of aggregate can successfully ameliorate the mechanical traits of the specimens but adding FA filler to the matrix with high-content aggregate is better. Reducing the proportion of aggregate can reduce the pore defects of specimens but using the appropriate quantity of FA filler is more effective than simply reducing the proportion of aggregate. Therefore, when improving the mechanical properties of sulfur cementitious materials, adding the appropriate amount of FA filler based on the CP aggregate matrix instead of blindly reducing the amount of aggregate is suggested.The supplement of the appropriate amount of FA filler can improve the gradation of aggregate and the compactness of the matrix and effectively remove air bubbles in the casting process, thus reducing the porosity of the specimen and enhancing its strength. FA particles play the position of crystal nuclei during the cooling and crystallization of sulfur. In the cooling and molding of specimens, FA particles, as the kernel of sulfur crystal development, can boost the sulfur crystal switch. Liquid sulfur accumulates, seriously changes close to the area of the FA particles, develops and extends around the particles, and subsequently bonds into a uniform matrix. Ceramic aggregate and fly ash filler can be used as weak interface transition zone of the WSCC, and sulfur can bond ceramic aggregate and fly ash filler to form a uniform and firm structure under an ideal covering condition.The dosage of sulfur and CP is closely related to the FS, CS, STS, FCR, and TCR of the specimen. The dosage of FA filler also has a certain degree of influence. When the mass ratio of sulfur: CP: FA is 1:2.7:0.3, the comprehensive mechanical properties of the WSCC is best. At this point, sulfur, CP, and FA can achieve the ideal bonding state. Sulfur can wrap all CP and FA particles, the mechanical engagement between particles is best, and the freshly mixed slurry in the construction process is easy to pour and vibrate. After the specimen is cooled and formed, minimal natural defects and bonding cracks are observed, and the mechanical properties of the specimen can be remarkably improved. The FS, CS, and STS of the specimen are 14.8, 86.2, and 6.8 MPa, respectively. The FCR and TCR are 0.172 and 0.079, respectively.

## Figures and Tables

**Figure 1 materials-16-01203-f001:**
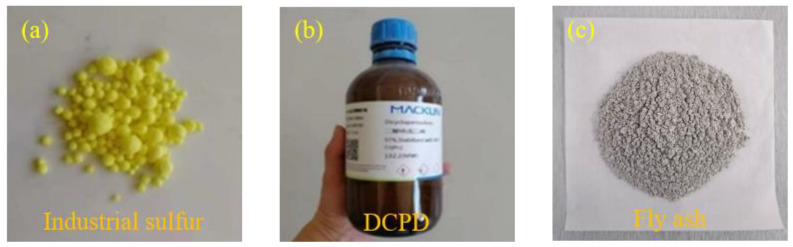
Industrial sulfur, modifier, and fly ash (**a**) Industrial sulfur; (**b**) DCPD; (**c**) Fly ash.

**Figure 2 materials-16-01203-f002:**
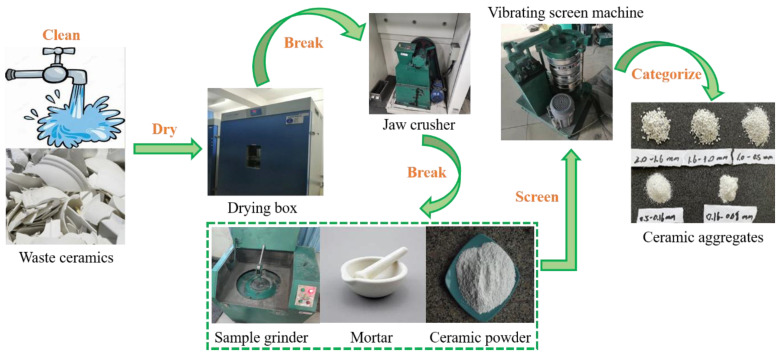
Preparation process of ceramic aggregates.

**Figure 3 materials-16-01203-f003:**
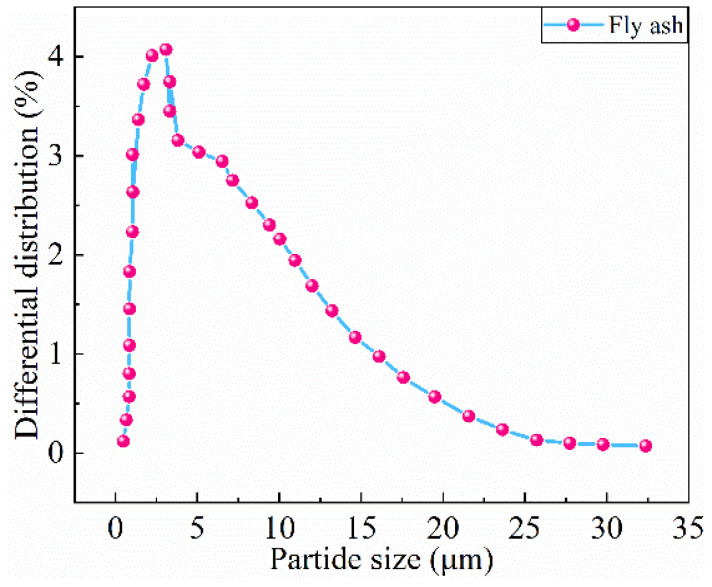
Particle size diagram of fly ash.

**Figure 4 materials-16-01203-f004:**
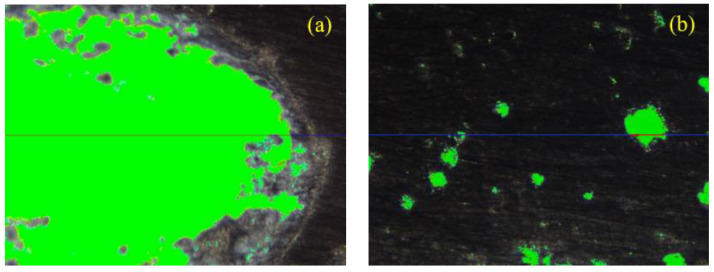
Principle of stomatal parameter test (**a**) the length of the line segment intercepted by the stomata; (**b**) the full length of the line in the sample.

**Figure 5 materials-16-01203-f005:**
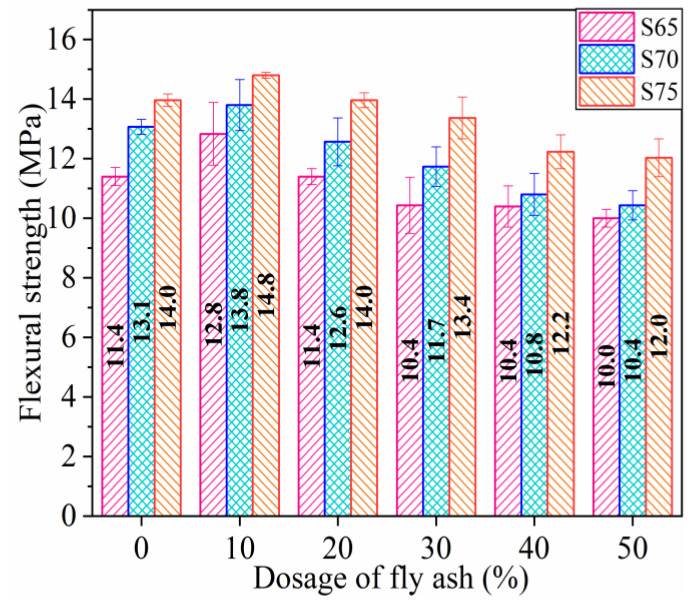
Flexural strength of specimens.

**Figure 6 materials-16-01203-f006:**
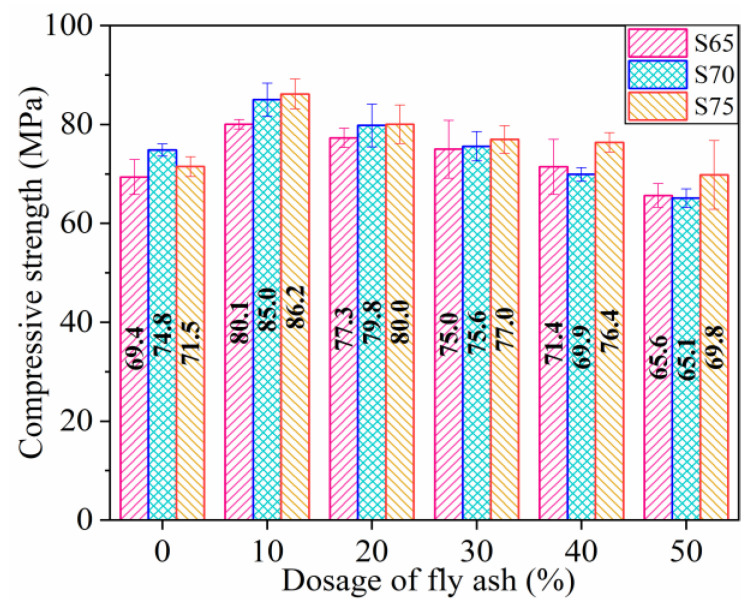
Compressive strength of specimens.

**Figure 7 materials-16-01203-f007:**
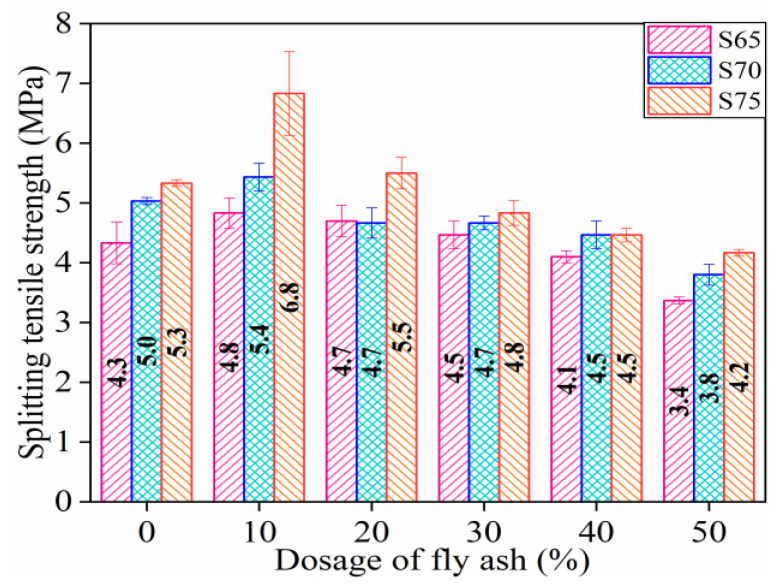
Splitting tensile strength of specimens.

**Figure 8 materials-16-01203-f008:**
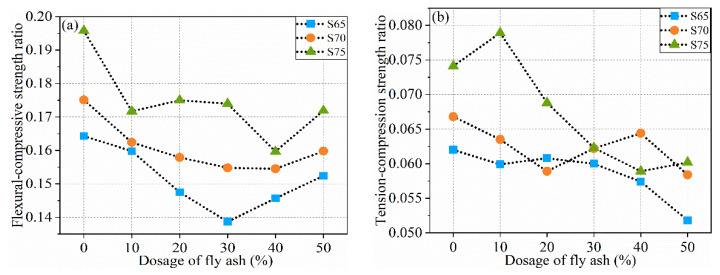
Flexural and the tension compression ratios of specimens (**a**) FCR; (**b**) TCR.

**Figure 9 materials-16-01203-f009:**

Influence of 0% and 10% fly ash on the appearance of specimens (**a**) S65F0 and S65F10; (**b**) S70F0 and S70F10; (**c**) S75F0 and S75F10.

**Figure 10 materials-16-01203-f010:**

Pore structure analysis of specimens (**a**) S70F0; (**b**) S70F10; (**c**) S75F0; (**d**) S75F10.

**Figure 11 materials-16-01203-f011:**
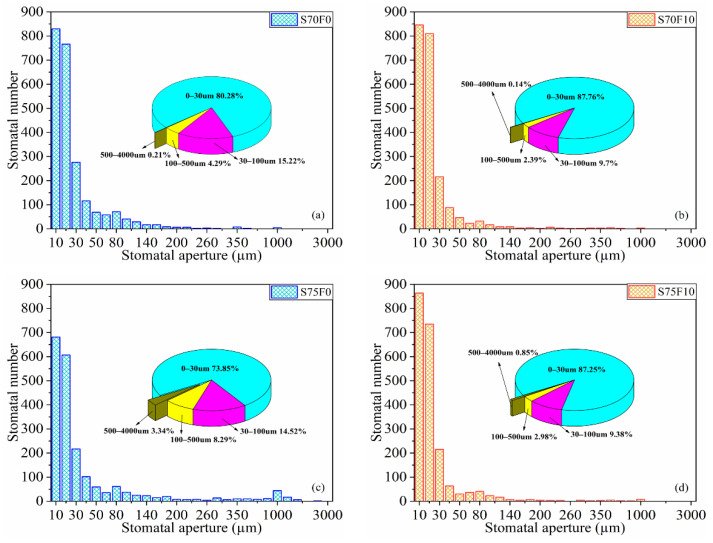
Distribution and proportion of the pore sizes of specimens (**a**) S70F0; (**b**) S70F10; (**c**) S75F0; (**d**) S75F10.

**Figure 12 materials-16-01203-f012:**
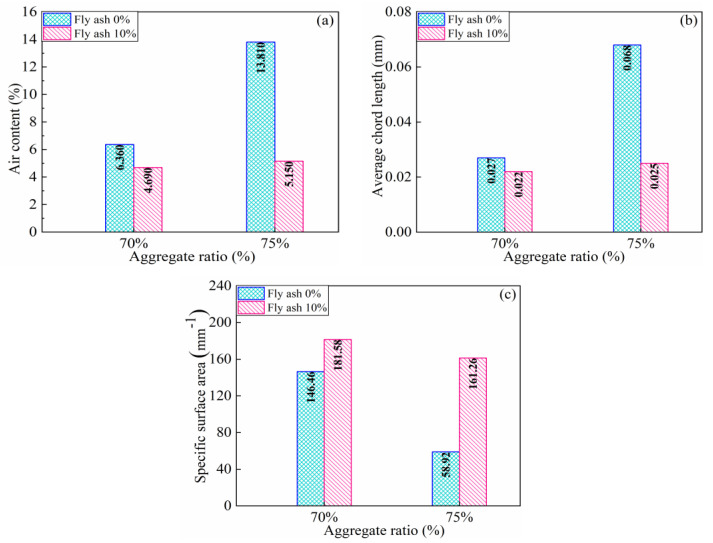
Pore structure of specimens (**a**) air content; (**b**) average chord length; (**c**) specific surface area.

**Figure 13 materials-16-01203-f013:**
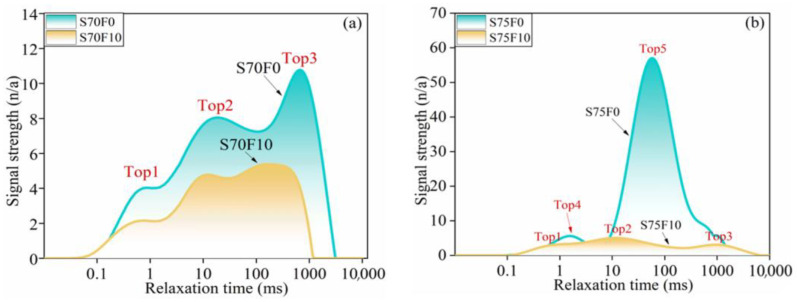
*T*_2_ spectrum of specimens (**a**) S70F0 and S70F10; (**b**) S75F0 and S75F10.

**Figure 14 materials-16-01203-f014:**
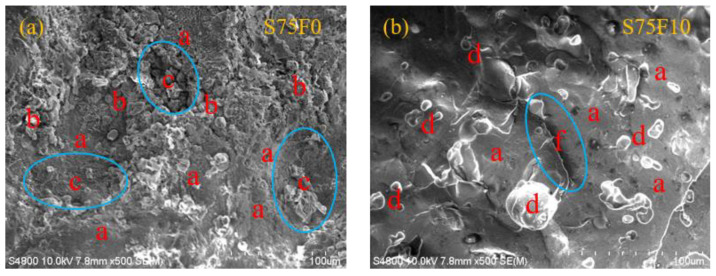
Micromorphology of specimens under SEM (400×) (**a**) S75F0; (**b**) S75F10: a = plate crystallization; b = chain crystallization; c = flaw; d = fly ash; f = crack.

**Figure 15 materials-16-01203-f015:**
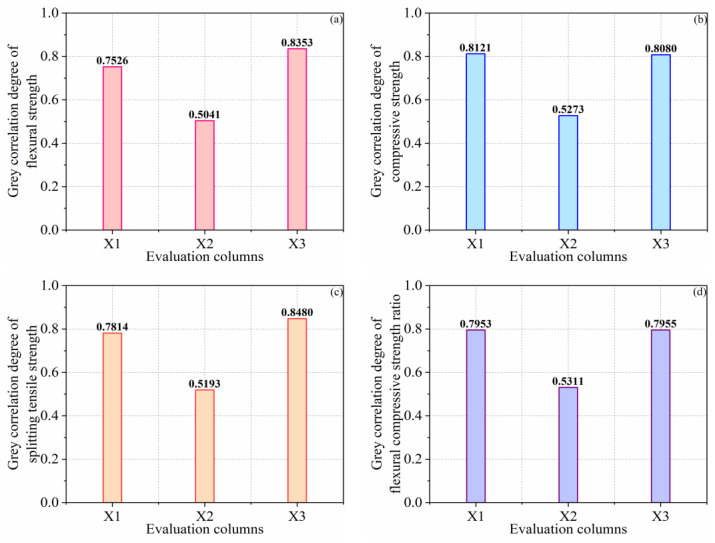
Gray correlation degree between influencing factors and reference indices (**a**) reference indices is FS; (**b**) reference indices is CS (**c**) reference indices is STS; (**d**) reference indices is FCR (**e**) reference indices is TCR.

**Figure 16 materials-16-01203-f016:**
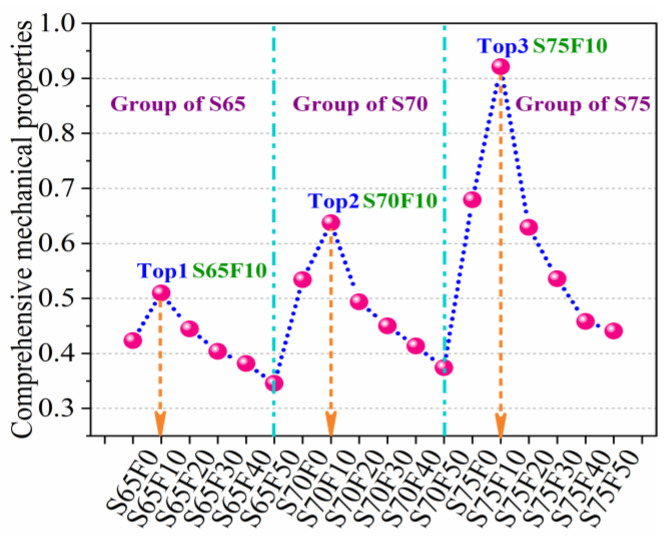
Comprehensive mechanical properties of specimens.

**Table 1 materials-16-01203-t001:** Particle size and its proportion.

Particle Size (mm)	0.08–0.16	0.16–0.5	0.5–1.0	1.0–1.6	1.6–2.0
Percentage (%)	13	20	34	26	7

**Table 2 materials-16-01203-t002:** Main physical performance of aggregates.

Aggregate Type	Origin	Apparent Density (kg/m^3^)	Sludge Content (%)
Discarded ceramics	Jingdezhen City	2460	0

**Table 3 materials-16-01203-t003:** Chemical constitution of ceramic powder.

Oxide	SiO_2_	Al_2_O_3_	CaO	Fe_2_O_3_	MgO	K_2_O	Na_2_O	Other
Composition	52.20	45.20	0.37	0.42	0.12	0.04	0.15	1.50

**Table 4 materials-16-01203-t004:** Chemical constitution of fly ash.

Oxide	SiO_2_	Al_2_O_3_	CaO	Fe_2_O_3_	MgO	K_2_O	Na_2_O	Other
Composition	40.28	18.15	18.08	8.56	2.43	1.76	1.31	2.26

**Table 5 materials-16-01203-t005:** Mix proportions of the tested specimens.

Group	Binder (%)	Aggregate and Filler (%)			Mass Ratio		
	Sulfur (%)	Total Percentage (%)	Ceramic Powder (%)	Fly Ash (%)	Sulfur	Ceramic Powder	Fly Ash
S65F0	35	65	100	0	1.0	1.9	0.0
S65F10	35	65	90	10	1.0	1.7	0.2
S65F20	35	65	80	20	1.0	1.5	0.4
S65F30	35	65	70	30	1.0	1.3	0.6
S65F40	35	65	60	40	1.0	1.1	0.7
S65F50	35	65	50	50	1.0	0.9	0.9
S70F0	30	70	100	0	1.0	2.3	0.0
S70F10	30	70	90	10	1.0	2.1	0.2
S70F20	30	70	80	20	1.0	1.9	0.5
S70F30	30	70	70	30	1.0	1.6	0.7
S70F40	30	70	60	40	1.0	1.4	0.9
S70F50	30	70	50	50	1.0	1.2	1.2
S75F0	25	75	100	0	1.0	3.0	0.0
S75F10	25	75	90	10	1.0	2.7	0.3
S75F20	25	75	80	20	1.0	2.4	0.6
S75F30	25	75	70	30	1.0	2.1	0.9
S75F40	25	75	60	40	1.0	1.8	1.2
S75F50	25	75	50	50	1.0	1.5	1.5

**Table 6 materials-16-01203-t006:** Results of mean processing.

Specimen	Evaluation Columns			Reference Columns				
	Sulfur	CP	FA	FS	CS	STS	FCR	TCR
	*X* _1_	*X* _2_	*X* _3_	*Y* _1_	*Y* _2_	*Y* _3_	*Y* _4_	*Y* _5_
S65F0	1.1667	0.0000	1.2381	0.9361	0.9250	0.9176	1.0125	0.9882
S65F10	1.1667	0.3714	1.1143	1.0511	1.0677	1.0235	0.9847	0.9548
S65F20	1.1667	0.7429	0.9905	0.9361	1.0308	0.9953	0.9089	0.9691
S65F30	1.1667	1.1143	0.8667	0.8540	1.0001	0.9459	0.8547	0.9563
S65F40	1.1667	1.4857	0.7429	0.8540	0.9525	0.8682	0.8978	0.9149
S65F50	1.1667	1.8571	0.6190	0.8212	0.8752	0.7129	0.9391	0.8256
S70F0	1.0000	0.0000	1.3333	1.0757	0.9979	1.0659	1.0790	1.0647
S70F10	1.0000	0.4000	1.2000	1.1332	1.1334	1.1506	1.0008	1.0121
S70F20	1.0000	0.8000	1.0667	1.0347	1.0641	0.9882	0.9730	0.9388
S70F30	1.0000	1.2000	0.9333	0.9608	1.0081	0.9882	0.9539	0.9914
S70F40	1.0000	1.6000	0.8000	0.8869	0.9321	0.9459	0.9521	1.0265
S70F50	1.0000	2.0000	0.6667	0.8540	0.8681	0.8047	0.9847	0.9308
S75F0	0.8333	0.0000	1.4286	1.1496	0.9534	1.1294	1.2066	1.1811
S75F10	0.8333	0.4286	1.2857	1.2153	1.1490	1.4471	1.0581	1.2576
S75F20	0.8333	0.8571	1.1429	1.1496	1.0672	1.1647	1.0784	1.0966
S75F30	0.8333	1.2857	1.0000	1.1004	1.0263	1.0235	1.0722	0.9930
S75F40	0.8333	1.7143	0.8571	1.0018	1.0183	0.9459	0.9841	0.9388
S75F50	0.8333	2.1429	0.7143	0.9854	0.9308	0.8824	1.0593	0.9595

## Data Availability

The data presented in this study are available on request from the corresponding author.

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
