# Peer review of "Investigation on the Mechanical Properties and Strengthening Mechanism of Solid-Waste–Sulfur-Based Cementitious Composites"

_materials, 2023, doi:10.3390/ma16031203_

Round 1

Reviewer 1 Report

The paper entitled “Investigation on the Mechanical Properties and Strengthening Mechanism of Solid Waste–Sulfur-Based Cementitious Composites” brings an interesting study about sustainable method for the enhancement of the mechanical characteristics of sulfur-based cementitious composite by introducing ceramic powder to replace natural aggregate completely and sulfur and fly ash filler for the preparation of waste–sulfur-based cementitious composites. Suggestions for improvements can be viewed in the attached file.

Author Response

The author is grateful to the reviewers and editors for their comprehensive and rigorous comments on this paper, which greatly improved our work. The author has carefully revised the paper according to the opinions of reviewers, and the words in blue is the revised content. Please refer to the attachment for details.

Reviewer 2 Report

This paper investigates the mechanical performance of cement mortar when the waste ceramic powder is added. The study shows promising results and interest in the concrete community. I have a few concerns before acceptance.

1.     Fly ash itself is a waste product and is considered either as partial cement replacement (as supplementary cementitious material) or complete cement replacement (flyash-based geopolymer). This paper proposed waste ceramic powder as a substitute for flyash. What are the significant beneficial?

2.     For particle size distribution, it is encouraged to plot rather than list in the table (referring to Table 1).

3.     In Figure 9, what is the physical meaning of stomatal aperture and stomatal number? The authors did not discuss these terms and how they are used in quantifying the pore sizes.

4.     The authors should add a section that discusses the grey scale analysis and entropy method.

5.     In Figure 10, the authors failed to explain how the terms air content, average chord length, and specific surface area are calculated. What is the average chord length?

6.     What is T2 spectrum curve? What is their significance? The authors should elaborate.

7.     What is natural crushi in line 77?

8.     The authors should detail how tensile and flexural tests are carried out in this paper.

9.     How is the workability with the addition of ceramic powder?

10.  What is the water-to-cement ratio in this study or the water-to-binder ratio since this study considered adding fly ash and ceramic powder? 

Author Response

(The authors gave the same response as above.)

Round 2

Reviewer 1 Report

The paper was revised according to the suggestions made. I believe it is fit for publication.

Reviewer 2 Report

The authors have addressed the concerns satisfactorily.